# Thymic Hyperplasias in Practice: Clinical Context, Histological Clues, and Management Implications

**DOI:** 10.3390/cancers18010084

**Published:** 2025-12-27

**Authors:** Stefan Porubsky

**Affiliations:** Institute of Pathology, University Medical Center of the Johannes Gutenberg University Mainz, Langenbeckstraße 1, 55131 Mainz, Germany; stefan.porubsky@unimedizin-mainz.de

**Keywords:** thymic hyperplasia, true thymic hyperplasia, rebound thymic hyperplasia, lymphofollicular thymic hyperplasia and lymphoepithelial-sialadenitis (LESA) like thymic hyperplasia

## Abstract

Thymic hyperplasia refers to several benign conditions in which the thymus becomes enlarged or develops histological changes. These conditions can look similar but differ greatly in who they affect, why they occur, and what they mean for patient care. This review brings together current knowledge about the main types of thymic hyperplasia to clarify how they can be distinguished under the microscope and in clinical practice. By improving the ability to differentiate between these conditions, this review article aims to support more accurate diagnoses, and guide appropriate follow-ups.

## 1. Introduction

Benign tumorous processes of the anterior mediastinum encompass a variety of diseases such as cysts, mesenchymal tumors (e.g., thymolipoma, lipofibroadenoma, vessel malformations), ectopic tissue (particularly thyroid and parathyroid), mature teratomas, and various forms of thymic hyperplasia [1,2,3,4].

The term “thymic hyperplasia” (TH) pertains to a diverse group of conditions characterized either by an enlargement of the thymic gland or by hyperplasia of lymphoid follicles, which typically are not present in significant numbers within a normal thymus [5,6].

Given the thymic gland’s specific ontogeny, its weight should be evaluated relative to the patient’s age. However, systematic studies on age-adjusted thymus weights are scarce and primarily found in older literature, such as [7,8,9,10] (Table 1 and Table 2). In patients aged 0 to 16 years, Young and Turnbull developed regression equations for the estimation of thymus weight based on age, stature, body weight, and heart weight, respectively (Table 3).

Radiologic measurement of the thymus size showed an age-dependent decrease in the transverse diameter: 1st year: 34.2 ± 10.5; 1–3 years: 35.7 ± 9.8; 4–6 years: 33.6 ± 8.9; 7–10 years: 29.0 ± 10.1; 11–14 years: 18.8 ± 9.0; 15–18 years: 19.0 ± 8.4 (mean ± SD [mm]), whereas the anterio-posterior size remained unchanged [12]. Instead of relying on thymic size assessment alone, the radiologic evaluation should focus on a multiparametric approach that includes tissue composition, attenuation characteristics, and enhancement behavior. Correlation with age-appropriate appearance and clinical context is essential. Magnetic-resonance-imaging (MRI) features such as signal intensity on T1-/T2-weighted images, diffusion characteristics and computed-tomography (CT) densitometry help differentiate fat-containing or involuted thymus from solid soft-tissue lesions. Assessment of homogeneity, lobulated contour, nodularity, cystic change, or calcifications provides further important morphologic clues [13,14]. Ultrasound including elastography is an alternative for assessing the thymic gland in children and adolescents [15].

This review aims to provide a comprehensive overview of the spectrum of thymic hyperplasias by synthesizing all relevant literature available in Medline up to November 2025 and the author’s own experience. It focuses on histopathologic, epidemiological and clinical characteristics of true thymic hyperplasia (TTH), rebound thymic hyperplasia (RTH), thymic follicular hyperplasia (TFH), and thymic hyperplasia with lymphoepithelial sialadenitis-like features (LESA-like TH) and intends to delineate their distinguishing features, discuss diagnostic pitfalls-particularly in biopsies and highlight the clinical implications.

## 2. Materials and Methods

This article is a narrative review of the current literature on thymic hyperplasias. A comprehensive search was performed on the Medline database. All articles published on humans in English, German, and French for which at least an abstract was available in Medline up to November 2025 were considered. The search terms included TTH (78 publications), RTH (68 publications), FTH (164 publications), and LESA-like TH (6 publications). Descriptive anatomical and radiological studies on the morphology of the thymus gland were included.

## 3. Results and Discussion

### 3.1. True Thymic Hyperplasia

TTH is defined as an increase in size and weight of the thymic gland while maintaining its age-appropriate architecture and lacking lymphoid follicles [5,16]. The organ weight exceeds the expected age-adjusted upper limit [7,16,17] (Table 1, Table 2 and Table 3). The condition is typically observed in infants and children, in whom the thymus may reach weights over 100 g and is very rare in adults [18,19].

On CT, TTH demonstrates a smoothly contoured thymus with homogeneous attenuation appropriate for patient age and without invasion, or associated lymphadenopathy [20]. MRI with chemical shift imaging shows signal loss on opposed-phase sequences due to preserved microscopic fat, a feature shared with other benign thymic hyperplasias but absent in most thymic neoplasms [21,22,23,24].

The etiology and molecular pathogenesis of TTH remain unknown; however, secondary forms have been documented in association with endocrine disorders such as acromegaly, hypopituitarism, Graves’ disease, and Addison’s disease [5]. The expansion of thymocytes is polyclonal. TTH may produce localized symptoms related to a mediastinal mass and can be further complicated by acute bleeding, which may be life-threatening [17,25]. In contrast to TFH and LESA-like thymic hyperplasia (discussed later), TTH is not associated with autoimmune disease.

Histologically, TTH exhibits a normal thymic architecture characterized by sharply demarcated cortical and medullary zones, the presence of Hassall’s corpuscles, thin interlobular septa, and an unremarkable capsule. Intervening adipose tissue is sparse. Features of involution and lymphoid follicles are absent (Figure 1).

The diagnosis of TTH is usually straightforward when analyzing whole-organ resections [26]. Type B1 thymomas can display an organoid pattern that resembles normal thymic tissue. The identification of a fibrous capsule and septa, along with medullary islands lacking Hassall’s corpuscles, supports the diagnosis of B1 thymoma [16]. It is noteworthy that the age distributions of TTH (primarily occurring in the first years of life) and thymomas (commonly found in adults and rarely in children) virtually do not overlap.

T-lymphoblastic lymphoma is characterized by an expansile proliferation of lymphoblasts, clonal T-cell receptor rearrangement, and frequent expression of LMO2 [27]. Differentiating TTH from RTH—another process with preserved thymic architecture—is generally straightforward when clinical information is accessible (see below). Furthermore, RTH predominantly occurs in adolescents and adults.

Given the specific differential diagnoses outlined above, establishing the correct diagnosis from a limited biopsy sample can be challenging, or even impossible. Therefore, the decision to perform a biopsy should be made with caution and take into account all relevant clinical and radiologic differential considerations. Finally, the possibility that the biopsy may have inadvertently sampled normal thymic tissue, unrelated to the pathological process, necessitates careful radiologic correlation.

Following thymectomy, TTH requires no specific long-term follow-up, as it represents a benign process without documented risk of recurrence or malignant transformation once complete excision and histological exclusion of the aforementioned differential diagnoses were achieved.

### 3.2. Rebound Thymic Hyperplasia

RTH retains a normal organ architecture, exhibiting only minimal or absent age-typical involutional alterations. The extent of enlargement is moderate, with thymic weights generally remaining below 100 g. The condition is typically asymptomatic and is most frequently identified incidentally in children and young adults following recovery from infectious diseases or subsequent to discontinuation of immunosuppressive therapy, radiotherapy, or chemotherapy. Additionally, it has been documented in patients with endocrine disorders [5,28,29]. The molecular pathogenesis is unknown. The expansion of lymphocytes is polyclonal. The radiological considerations do not differ from TTH (see above).

Histologically, the thymic architecture is preserved, with clearly identifiable cortical and medullary compartments (Figure 2). Hassall’s corpuscles are present, albeit at a lower frequency than in TTH. The lobules are separated by adipose tissue; however, the extent of involution is surprisingly limited compared with age-matched, non-affected thymi.

When appropriate clinical history is available, the diagnosis of RTH is generally straightforward. The considerations for differential diagnosis and biopsy largely overlap with those of TTH (see above).

The clinical and prognostic implications of RTH remain undetermined. Some publications suggest that the occurrence of RTH could represent a favorable prognostic indicator in neoplastic diseases [30,31]. In about half of the patients, spontaneous regressions have been described over a couple of years [32]. Following thymectomy, RTH as such does not require specific long-term surveillance, as it represents a reactive and self-limited process. A follow-up may be necessary for the underlying trigger (e.g., recovery after chemotherapy, immunosuppression, severe illness, or endocrine imbalance).

### 3.3. Thymic Follicular Hyperplasia

TFH is characterized by the presence of lymphoid follicles in more than one-third of the thymic lobules. The term “hyperplasia” refers to the increased number of hyperplastic lymphoid follicles, whereas the thymic gland itself generally shows no significant or only minimal enlargement and is typically not associated with mass-related symptoms [6]. The link between TFH and myasthenia gravis (MG) was first recognized by Laquer and Weigert in 1901 [33].

TFH is frequently observed in patients with early-onset myasthenia gravis (EOMG) who show anti–acetylcholine receptor (AChR) autoantibodies and is rather uncommon in MG patients with alternative serological profiles, such as antibodies against titin, ryanodine receptor, or muscle-specific tyrosine kinase (MuSK). TFH is not limited to EOMG and may also be present in patients with other autoimmune disorders or even in asymptomatic individuals [34,35,36].

In EOMG, AChR-reactive T cells re-enter the thymus and are activated by medullary thymic epithelial cells expressing MHC/AChR complexes. These T cells stimulate thymic B cells to produce antibodies, promoting germinal center formation and high-affinity anti-AChR antibody production. Functionally impaired regulatory T cells may sustain autoimmunity, though the exact origin of thymic lymphoid follicles and the prevalence of AChR-reactive T cells in healthy individuals remain unknown [35,37].

TFH can be distinguished radiologically from other anterior mediastinal processes by preservation of normal thymic architecture and the presence of fat. On CT, TFH typically appears as diffuse, symmetric thymic enlargement with smooth margins and without invasive features or lymphadenopathy, as opposed to thymic epithelial tumors or lymphomas, which usually present as bulky masses. MRI with chemical shift imaging is the most reliable modality, as TFH demonstrates signal loss on opposed-phase images due to intermixed fat, a finding generally absent in thymic neoplasms. Fluorodeoxyglucose-Positron Emission Tomographymay show mild to moderate uptake but lacks specificity [20,21,22,23].

Histologically, TFH is characterised by the presence of lymphoid follicles within the medulla and perivascular spaces (Figure 3). These follicles can be highlighted by immunohistochemical staining for CD20 and CD23 (or CD21) and contain the typical composition of CD10- and Bcl6-positive, Bcl2-negative germinal center lymphocytes. The follicular structures lack the medullary epithelial meshwork, which is displaced to the periphery of the affected lobules. The cortical rim appears thinned or disrupted, a feature that can be demonstrated via TdT immunohistochemistry. Thymic lobules unaffected by follicular hyperplasia retain a normal architecture. It is noteworthy that prior administration of immunosuppressive therapy, frequently employed in the treatment of MG, may induce significant thymic atrophy, especially within the cortical compartment.

In biopsy specimens, establishing the diagnosis of TFH may be challenging, as sampling error can lead to omission of affected lobules. Conversely, isolated lymphoid follicles may be encountered in healthy individuals and may not necessarily indicate disease, particularly when fewer than one-third of thymic lobules are involved. Their detection may therefore be nonspecific and unrelated to a neoplastic process that may have been missed by the biopsy. Detailed information regarding MG serology is diagnostically valuable, as antibodies against titin, ryanodine receptor, and MuSK are less common in TFH and should raise suspicion for thymoma-associated or late-onset MG [34]. In a typical setting of EOMG, biopsy has a very limited role and may only be warranted if the radiologic findings suggest important differential diagnoses.

Micronodular thymoma (or carcinoma) with lymphoid stroma also exhibits prominent lymphoid follicles; however, it can be readily distinguished from TFH by its compact trabecular epithelial component and by the absence of lymphoepithelial lesions, or Hassall’s corpuscles. Unlike TFH, micronodular thymoma with lymphoid stroma demonstrates a predilectional localisation of immature TdT-positive T cells at the epithelial–lymphoid interface, with limited infiltration into the epithelial trabeculae. Micronodular carcinoma with lymphoid stroma lacks immature thymocytes.

Follicles with atypical architecture should prompt consideration of a lymphoma, particularly mucosa-associated lymphoid tissue (MALT) lymphoma, and warrant further hematopathological assessment. Castleman disease might be considered when characteristic small, onion-skin–like follicles and hyalinised vessels are observed. Differentiation from extensively inflamed infiltrates of extrathymic carcinomas or germ cell tumors is achieved by organ-specific immunohistochemical markers. The differential diagnosis between TFH and LESA-like TH is discussed below.

Thymectomy is a standard treatment for AChR antibody–positive myasthenia gravis and is preferably performed early after disease onset. While removal of the thymus can reduce thymic autoantigen presentation and pathogenic immune cells and leads to clinical improvement in about half of patients, complete remission is uncommon. Persisting autoreactive B cells and plasma cells likely contribute to ongoing autoantibody production and poorer outcomes [38].

### 3.4. Thymic Hyperplasia with Lymphoepithelial Sialadenitis (LESA)-like Features

The term “thymic hyperplasia with lymphoepithelial sialadenitis–like features (LESA-like TH)” was introduced in 2012 by Weissferdt and Moran, who described 4 patients with a tumorous proliferation composed of thymic epithelial cells, lymphoid follicles, and cystic changes [39]. In line with subsequent case reports [40,41,42,43], the so far largest cohort of 36 patients showed a broad size range (1–21 cm), a mean age at presentation of 52 years (range, 32–80 years) and a male to female ration of 1.4:1 [44]. The lesion represents mostly an incidental finding or manifests through compression of the heart, great vessels, or lungs in singular cases. The etiology and pathogenesis are unknown. Radiologic studies addressing diagnostic criteria and the differential diagnosis of LESA-like thymic hyperplasia are currently lacking. The lesion may exhibit cystic features. Notably, a recent study reported increased fluorodeoxyglucose avidity in a patient with LESA-like thymic hyperplasia [43].

The aforementioned cohort of Porubsky et al. indicated that 33% of patients experienced non-myasthenic autoimmune diseases, including systemic lupus erythematosus, rheumatoid arthritis, scleroderma, Sjögren syndrome, pure red cell aplasia, or Graves’ disease [44]. Additionally, 14% of these patients had an associated lymphoma-most commonly MALT lymphoma, with one case of diffuse large B-cell lymphoma-indicating that LESA-like TH is not simply an exaggerated form of TFH but rather a distinct entity.

Histologically, LESA-like TH is characterized by abundant lymphoid follicles that may impart a lymph node-like appearance and obscure the underlying thymic architecture at low magnification (Figure 4). In the periphery, uninvolved lobes or lobes with mild follicular hyperplasia may persist. Higher magnification or cytokeratin immunohistochemistry reveal a hyperplastic epithelial meshwork with prominent lymphoepithelial lesions. Hassall’s corpuscles are hyperplastic and demonstrate micro- and macro-cystic alterations, which in some cases are sufficiently pronounced to mimic a cystic lesion radiologically. The hyperplastic medullary epithelium and lymphoid follicles form interdigitating yet sharply demarcated geographic patterns. Cortical structures containing immature TdT-positive thymocytes are typically absent in the affected regions. As a significant part of the cases was linked to a lymphoma, an aberrant follicular architecture should elicit further hematopathological investigations including immunohistochemical studies and, in uncertain cases, B cell receptor rearrangement analysis (Figure 5).

The differential diagnosis largely overlaps with that of TFH. Distinction between TFH and LESA-like TH can be challenging and should take into account that prominent lymphoepithelial lesions, together with epithelial hyperplasia and hyperplasia of Hassall’s corpuscles, represent the most characteristic features of LESA-like TH. In addition, at least incipient microcyst formation within hyperplastic Hassall’s corpuscles appears to be typical of LESA-like TH (Figure 4G). Larger cysts containing eosinophilic material and cholesterol crystals (Figure 4B) are not required for the diagnosis but, when present, support LESA-like TH. Although lymphoid follicles may be numerous in TFH, their density is typically significantly higher in LESA-like TH. A “lymph node–like” architecture is not a conditio sine qua non; however, when present, it favors a diagnosis of LESA-like TH. In both TFH and LESA-like TH, loss of cortical structures parallels the development of lymphoid follicles. Accordingly, in LESA-like TH, areas of exuberant follicle formation are largely devoid of cortex and immature TdT-expressing lymphocytes. In patients with TFH-associated EOMG, prior immunosuppressive therapy may also result in marked cortical atrophy; therefore, cortical atrophy alone cannot be used as a distinguishing criterion. Finally, a clinical history of MG appears to be very uncommon in patients with LESA-like TH, whereas a substantial proportion of these patients seem to suffer from non-myasthenic autoimmune disorders.

When cystic change is prominent, mediastinal cysts or inflammatory lesions with secondary cystic change may enter the differential diagnosis, but these processes can be readily distinguished by the histopathological features described above.

Biopsy is generally not indicated for LESA-like TH, except in atypical cases, relevant radiologic differential diagnoses or in patients with a high surgical risk.

Given the association of LESA-like TH with autoimmune diseases and B-cell lymphoproliferative disorders, clinical follow-up should be considered despite the lack of standardized guidelines. A reasonable approach includes baseline clinical assessment with targeted hematologic, rheumatologic, and laboratory evaluation at the time of diagnosis. Patient education and regular clinical follow-up (e.g., annually), with particular attention to B symptoms, new mediastinal complaints, lymphadenopathy, and emerging immunologic abnormalities, appear appropriate.

## 4. Conclusions

The spectrum of TH encompasses biologically and clinically distinct entities that range from benign physiological or reactive processes to lesions requiring diagnostic work-up due to their association with autoimmune disease or lymphoma (Table 4 and Figure 6). TTH and RTH share preservation of normal thymic architecture, but differ in epidemiology and clinical context: TTH primarily affects infants and young children and may present with mass-related symptoms, whereas RTH typically arises in adolescents and young adults recovering from stressors such as infection, chemotherapy, or endocrine disturbances. Both entities lack an intrinsic association with autoimmunity and are defined histologically by intact corticomedullary organization, preserved Hassall’s corpuscles, and absence of lymphoid follicles.

In contrast, TFH and LESA-like TH represent immunologically driven processes with increasing architectural distortion. TFH is strongly associated with EOMG and is characterized by medullary lymphoid follicles. LESA-like TH, although also follicular in architecture, constitutes a separate entity defined by additional epithelial hyperplasia, prominent lymphoepithelial lesions, cystic change, and severe cortical atrophy. Its association with non-myasthenic autoimmune diseases and thymic MALT lymphoma underscores the need for hematologic and rheumatologic evaluation once diagnosed.

Together, these entities illustrate the spectrum of thymic hyperplasias and highlight the importance of integrating clinical context, serology, and detailed histopathological assessment for accurate classification.

## Figures and Tables

**Figure 1 cancers-18-00084-f001:**
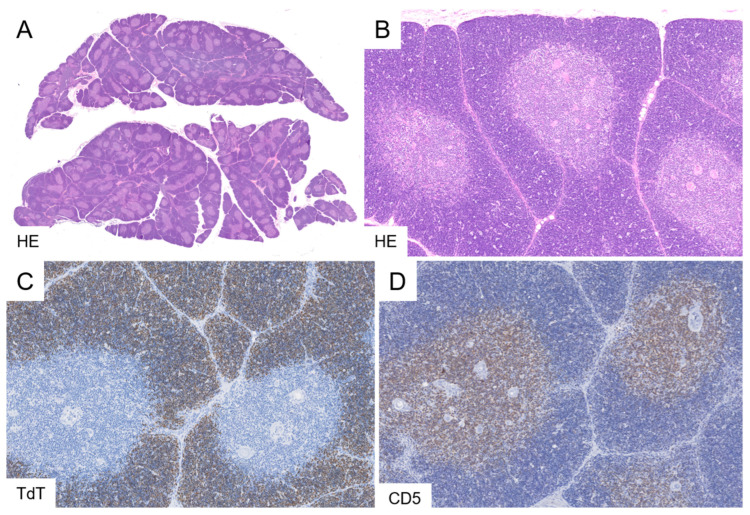
True thymic hyperplasia. A case of a 6-month-old infant with thymic enlargement displaying an age-appropriate organ architecture without intervening adipose tissue (**A**). The lobules are densely arranged and separated by delicate septa, with well-developed cortical and medullary compartments including Hassall’s corpuscles. The absence of lymphoid follicles is noted (**B**). Immunohistochemistry demonstrates immature TdT-positive, CD5-low thymocytes localized to the cortex, and mature TdT-negative, CD5-high thymocytes confined to the medulla (**C**,**D**). Original magnifications: (**A**), 2×; (**B**), 5×; (**C**,**D**), 10×.

**Figure 2 cancers-18-00084-f002:**
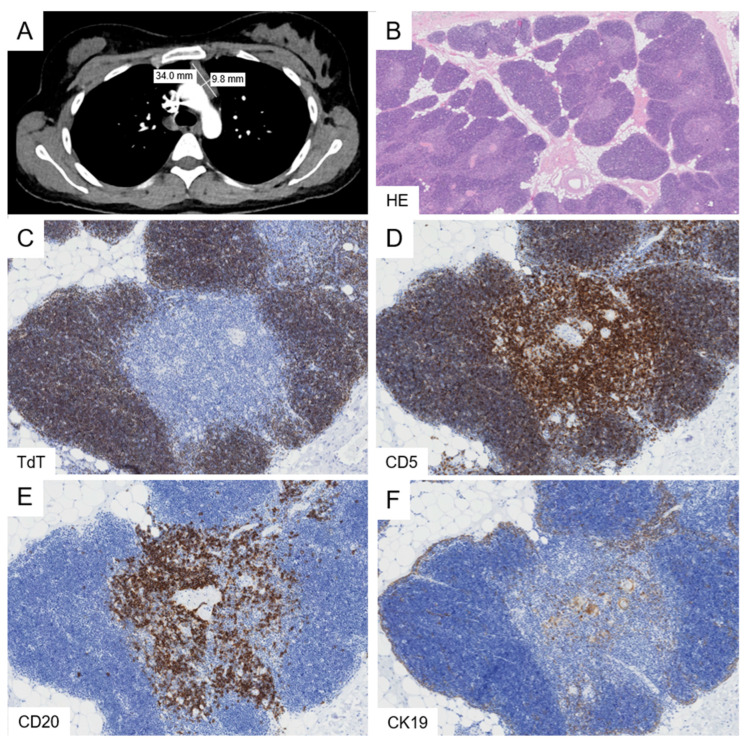
Rebound thymic hyperplasia. A case of a 25-year-old female patient with a history of respiratory infection presents with an anterior mediastinal mass measuring up to 3.4 cm on CT imaging (**A**). Histological examination reveals preserved thymic architecture with minimal involutional changes (**B**). Immunohistochemical stainings for TdT, CD5, CD20, and CK19 highlight intact cortical and medullary compartments without interspersed lymphoid follicles (**C**–**F**). Original magnifications: (**B**), 3×; (**C**–**F**), 10×.

**Figure 3 cancers-18-00084-f003:**
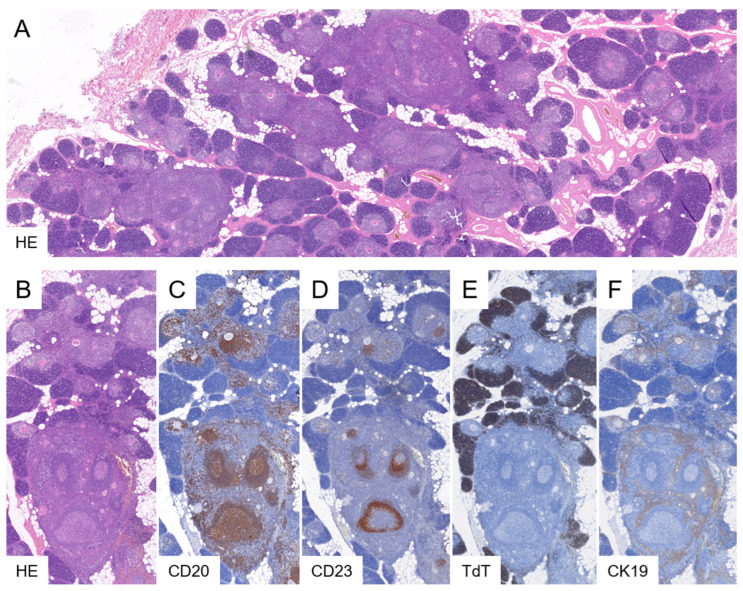
Thymic follicular hyperplasia. In TFH, the normal thymic architecture is disrupted by expanding lymphoid follicles involving subsets of lobules (**A**). These follicles, containing follicular dendritic cell networks, are highlighted by CD20 and CD23 immunostaining ((**B**–**D**), lower part). Affected lobules exhibit thinning of the TdT-positive cortex and distortion of the cytokeratin-positive epithelial meshwork ((**E**,**F**), lower part). In contrast, unaffected lobes seen in the upper parts of panels (**B**–**F**) retain typical histology with medullary CD20-positive B cells, cortical TdT-positive thymocytes, and a fine cytokeratin-positive epithelial network. CD23 expression is limited to scattered medullary B cells whereas CD23-positive follicular dendritic cells are absent in these regions ((**B**–**F**), upper part). Original magnifications: (**A**), 2×; (**B**–**F**), 3×.

**Figure 4 cancers-18-00084-f004:**
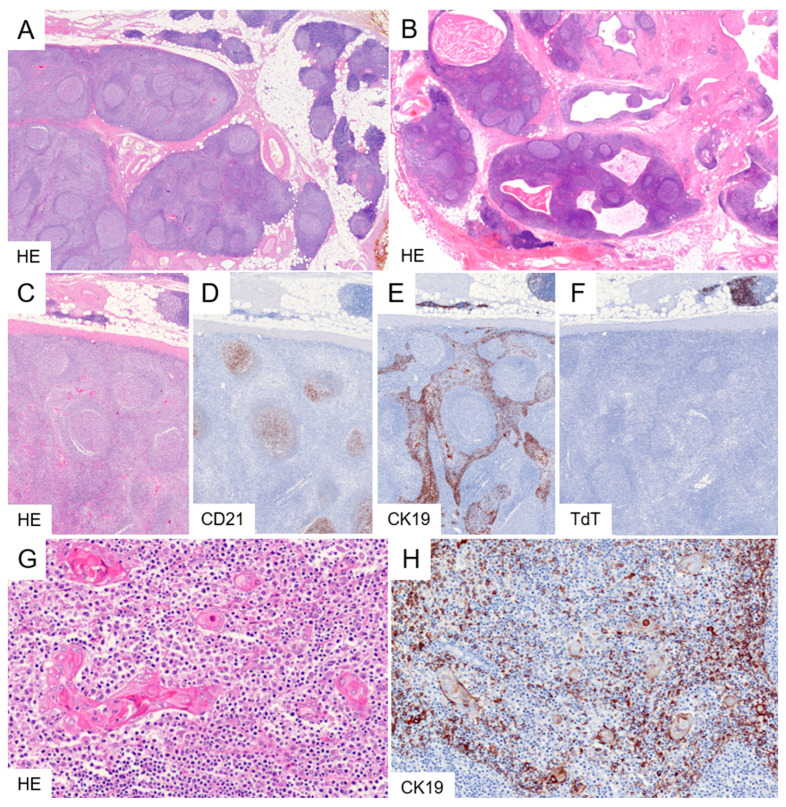
LESA-like thymic hyperplasia. At low magnification, LESA-like TH is characterized by numerous lymphoid follicles that obscure the native thymic architecture, along with cystic structures containing eosinophilic debris and cholesterol clefts. Residual, uninvolved thymic tissue is present at the periphery (**A**,**B**). CD21 immunostaining highlights follicular dendritic cells within lymphoid follicles abutting hyperplastic epithelium, forming sharply demarcated geographic epithelial–lymphoid interfaces (**C**–**E**). Affected regions lack immature TdT-positive thymocytes, in contrast to residual unaffected lobules identified in the upper part (**F**). Higher magnification and cytokeratin staining reveal epithelial hyperplasia, prominent lymphoepithelial lesions, and numerous Hassall’s corpuscles undergoing cystic transformation (**G**,**H**). Original magnifications: (**A**,**B**), 2×; (**C**–**F**), 4×; (**G**,**H**), 20×.

**Figure 5 cancers-18-00084-f005:**
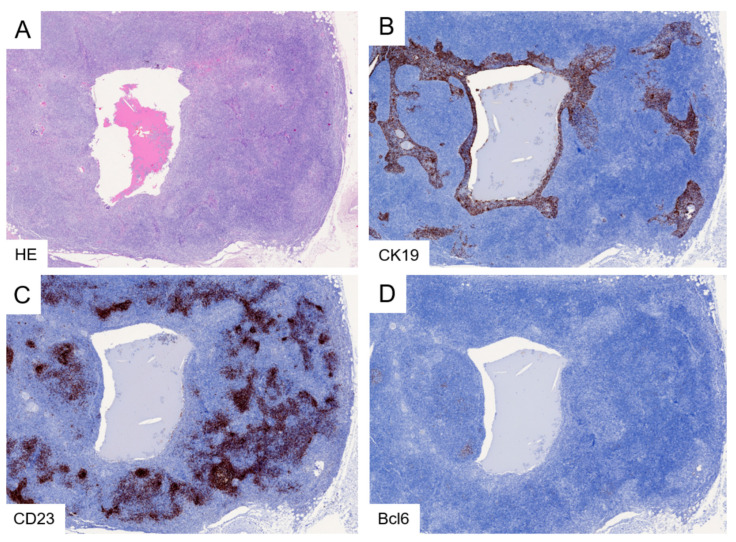
MALT lymphoma in LESA-like thymic hyperplasia. In a manner akin to LESA-like TH, the proliferating lymphoid tissue and hyperplastic epithelium establish complementary geographic patterns, which are accompanied by prominent cystic changes in this case (**A**,**B**). However, the characteristic lymphoid follicles are replaced by irregular networks of CD23-expressing follicular dendritic cells (**C**) colonized by Bcl6-negative lymphocytes (**D**). Original magnification: (**A**–**D**), 5×.

**Figure 6 cancers-18-00084-f006:**
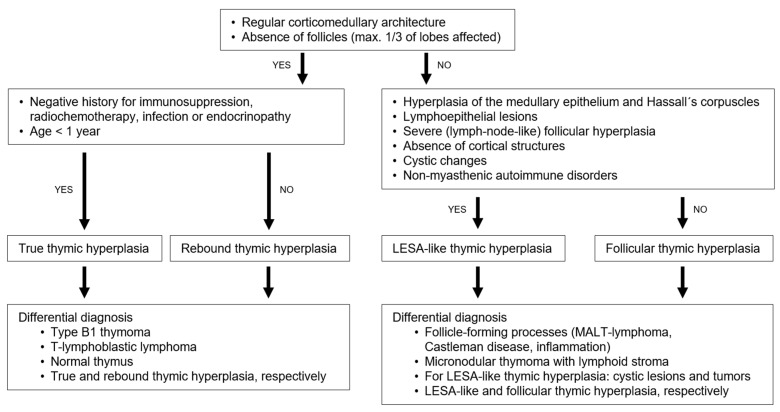
Conceptual diagnostic algorithm for thymic hyperplasia.

**Table 1 cancers-18-00084-t001:** Mean weight and standard deviation of the thymus at ages under 16 years [9,10,11].

	Full-Time Fetuses	<1 Year	1–6 Years	6–11 Years	11–16 Years
Mean [g]	14.68	18.89	23.71	29.18	32.13
SD [g]	8.11	10.12	9.14	10.71	12.22
n	104	134	183	136	95

**Table 2 cancers-18-00084-t002:** Mean weight and standard deviation of the thymus at ages over 16 years according to [9,11].

	16–21 Years	21–26 Years	26–31 Years	31–36 Years	36–46 Years	46–56 Years	56–66 Years	>66 Years
Mean [g]	22.49	19.24	16.29	16.83	15.70	15.55	13.34	10.51
SD [g]	10.64	10.63	8.61	8.70	8.34	10.20	8.27	7.19
n	53	53	39	30	37	38	28	21

**Table 3 cancers-18-00084-t003:** Regression equations for the estimation of the thymus weight according to Young and Turnbull [9].

Males	Females
y=0.290w+0.076h - 0.739a+19.325	y=0.478w+0.005h - 0.404a+19.811
Partial standard deviation (σy.wha) = 12.53	Partial standard deviation (σy.wha) = 12.01
Mean error of prediction = 9.7	Mean error of prediction = 8.9
Theoretical mean error = 10.0	Theoretical mean error = 9.6

y = thymus weight [g], w = body weight [kg], h = body height [cm], a = age [years].

**Table 4 cancers-18-00084-t004:** Summary of entities with thymic hyperplasia.

Entity	Epidemiology	Clinical Features	Histology	Differential Diagnosis
True Thymic Hyperplasia	Infants and children	Thymic enlargement without autoimmune disease	Preserved architecture, no lymphoid follicles	B1 thymoma, T-lymphoblastic lymphoma, RTH, normal thymus
Rebound Thymic Hyperplasia	Children and younger adults	After recovery from radio/chemotherapy, immunosuppression, infection or in endocrine disease	Preserved architecture with mild age-typical atrophy, no lymphoid follicles	B1 thymoma, T-lymphoblastic lymphoma, TTH, normal thymus
Thymic Follicular Hyperplasia	Younger adults	In autoimmune settings, particularly myasthenia gravis with anti-AChR antibodies	Medullary and perivascular lymphoid follicles with cortical thinning and epithelial network disruption	Micronodular thymoma with lymphoid stroma, LESA-like TH, reactive follicle-forming processes
LESA-like Thymic Hyperplasia	Adults	May be associated non-myasthenic autoimmune disease and MALT lymphoma	Severe architectural distortion with numerous lymphoid follicles, hyperplasia of the epithelium and Hassal’s corpuscles, cystic changes and absence of cortex	TFH, thymic MALT lymphoma, cystic mediastinal and thymic lesions

## Data Availability

Not applicable.

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
