# Peer review of "Thymic Hyperplasias in Practice: Clinical Context, Histological Clues, and Management Implications"

_cancers, 2025, doi:10.3390/cancers18010084_

Round 1

Reviewer 1 Report

Comments and Suggestions for Authors

Manuscript Title: Thymic Hyperplasias in Practice: Clinical Context, Histological  Clues, and Management Implications

This review is particularly valuable in highlighting LESA-like thymic hyperplasia as a distinct entity with strong associations to non-myasthenic autoimmune disease and thymic MALT lymphoma, which is still under-recognized in routine practice. The inclusion of illustrative tables and high-quality histology/immunohistochemistry images also has clear educational value for both thoracic pathologists and clinicians.

  • The Methods section currently only states that a comprehensive Medline search up to November 2025 was performed with a list of key terms, but there are no details on inclusion/exclusion criteria, study types, or how overlapping case series and reviews were handled.
  • The review presents classic autopsy data and regression equations (Tables 1–3), but it remains somewhat unclear how the author recommends using these in day-to-day diagnostic practice where weights may be unknown or only grossly estimated.
  • The text nicely describes the features of TTH, RTH, TFH, and LESA-like TH, but much of it is descriptive rather than criterion-based.
  • While the title promises “Management Implications”, the discussion of what exactly clinicians should do after a given diagnosis is still relatively brief.
  • The manuscript correctly states that LESA-like TH is not just an extreme form of TFH, but an independent entity strongly linked to systemic autoimmunity and lymphoma. Please consider adding a comparative table or short paragraph focusing just on the borderline scenarios: increasing follicular density, emerging lymphoepithelial lesions, extent of cortical loss, degree of cystic change, and correlation with serology or clinical phenotype.
  • In what situations do you recommend IGH rearrangement analysis to rule out MALT lymphoma within LESA-like TH?
  • A short integrative subsection summarising what is known (and unknown) about prognosis and outcomes for each entity would be valuable for clinicians (e.g., recurrence risk, progression to lymphoma, relevance to long-term MG control).
  • Overall the English is clear, but a careful proof-read would improve polish.
  • The histology/immunohistochemistry figures (Figures 1–5) are visually clear but the legends are very descriptive without specifying magnification or scale bars. Please include, for each panel, the objective magnification (e.g. “H&E, ×40/×100/×200”) and/or a scale bar. Ensure that all panel labels are explicitly described in the legend so the reader can easily map.
  • The Abbreviations list is helpful; please check that all abbreviations (e.g., AChR, MuSK, MALT) are defined at first use in the text and align exactly with the list at the end.

Author Response

Reviewer #1

This review is particularly valuable in highlighting LESA-like thymic hyperplasia as a distinct entity with strong associations to non-myasthenic autoimmune disease and thymic MALT lymphoma, which is still under-recognized in routine practice. The inclusion of illustrative tables and high-quality histology/immunohistochemistry images also has clear educational value for both thoracic pathologists and clinicians.

The Methods section currently only states that a comprehensive Medline search up to November 2025 was performed with a list of key terms, but there are no details on inclusion/exclusion criteria, study types, or how overlapping case series and reviews were handled.

Thank you for this suggestion. This article is a narrative review. The number of publication on these disease forms is really limited allowing a comprehensive search. Now, I have precised this in the Methods section in the following way: All articles published on humans in English, German and French for which at least an abstract was available in Medline up to November 2025 were considered. The search terms included true thymic hyperplasia (78 publications), rebound thymic hyperplasia (68 publications), follicular thymic hyperplasia (164 publications), and lymphoepithelial-sialadenitis (LESA)-like thymic hyperplasia (6 publications). Descriptive anatomical and radiological studies on the morphology of the thymus gland were included

The review presents classic autopsy data and regression equations (Tables 1–3), but it remains somewhat unclear how the author recommends using these in day-to-day diagnostic practice where weights may be unknown or only grossly estimated.

In the revised manuscript, I present age-stratified CT reference values for thymic dimensions in children and adolescents. Detailed analyses of thymic parameters - including anteroposterior and transverse diameters as well as lobar measurements - in pediatric and adult populations have been previously reported by Colak et al. and Araki et al. However, the thymic diameters reported in adults by Araki et al. are larger than those described by Colak et al. in children and adolescents. This discrepancy underscores the need for a highly precise and standardized approach when estimating thymic size and highlights the potential for inter-institutional variability. Consequently, the revised manuscript further emphasizes that clinical imaging assessment of the thymus should not rely on size criteria alone. Instead, a multiparametric radiologic approach is recommended, integrating tissue composition, CT attenuation, internal morphology, enhancement patterns, and correlation with patient age and clinical context, with ultrasound and MRI serving as complementary imaging modalities.

The text nicely describes the features of TTH, RTH, TFH, and LESA-like TH, but much of it is descriptive rather than criterion-based.

I fully agree with the reviewer that in diagnostic disciplines such as pathology, the establishment of rigorous, criterion-based diagnoses is highly desirable for a clear distinction of entities. However, the field of thymic hyperplasias is currently limited by the very small number of published cases. For example, the total number of reported LESA-like thymic hyperplasias to date is only 44. Likewise, the available literature on rebound and true thymic hyperplasia largely consists of individual case reports or small case series rather than systematic studies. Even the diagnostic threshold commonly used for thymic follicular hyperplasia - namely, the presence of follicles in more than one third of thymic lobes - should be regarded as an experience-based approximation rather than a criterion derived from robust statistical analysis. In practical terms, this implies that a diagnosis of thymic follicular hyperplasia would not reasonably be excluded in an AChR-positive early onset myasthenia gravis patient if follicles were present in, for example, only one quarter of the lobes examined. In this context, I hope that the diagnostic algorithm now presented in Figure 6 will improve practical clarity and support diagnostic decision-making. Nevertheless, I acknowledge that the current state of knowledge does not yet permit the formulation of truly evidence-based diagnostic criteria for these entities.

While the title promises “Management Implications”, the discussion of what exactly clinicians should do after a given diagnosis is still relatively brief.

In the revised manuscript, I have added a statement on management implications for each of these four entities. For TTH and RTH, however, these are rather short notices that no diagnosis-specific procedures are needed once a complete excision and histological diagnosis have been achieved. In LESA-like TH, the pathologists, clinicians and the patient should be aware of the described association, but it remains a yet unresolved question, how far the follow-up diagnostic procedures shall go in a lymphoma-negative LESA-like TH. In MG-associated TFH, a substantial proportion of the patients show an improvement in the symptoms, however, a complete curation is rather unlikely. Thus, most of them will require an ongoing neurologic follow-up.

The manuscript correctly states that LESA-like TH is not just an extreme form of TFH, but an independent entity strongly linked to systemic autoimmunity and lymphoma. Please consider adding a comparative table or short paragraph focusing just on the borderline scenarios: increasing follicular density, emerging lymphoepithelial lesions, extent of cortical loss, degree of cystic change, and correlation with serology or clinical phenotype.

In the revised manuscript, I have added a paragraph in the LESA-like TH section delineating the most typical and optional findings that should help to distinguish these two entities in practical terms.

In what situations do you recommend IGH rearrangement analysis to rule out MALT lymphoma within LESA-like TH?

Thank you for this remark. I have now rephrased the corresponding passage. I recommend to follow the general pathways of hematopathologic diagnostics starting with an assessment of the architecture of the lymphatic tissue using conventional microscopy and implementing further immunohistochemical studies on demand. If both show a physiological picture, I would restrain from further molecular studies in general. Similarly, if a lymphoma diagnosis is evident in terms of morphology and immunohistochemistry (e.g. with light chain restriction in a secretory differentiated MALT lymphoma), IGH rearrangement study is of no additional value. Thus, IGH rearrangement analysis shall be performed as a source of additional information in difficult cases that show unclear findings in conventional morphology and immunohistochemistry. 

A short integrative subsection summarising what is known (and unknown) about prognosis and outcomes for each entity would be valuable for clinicians (e.g., recurrence risk, progression to lymphoma, relevance to long-term MG control).

Whenever possible, this information has been supplemented, however especially in LESA-like TH, the experience is very limited.

Overall the English is clear, but a careful proof-read would improve polish.

The manuscript has been now thoroughly proof-red after performing the revision.

The histology/immunohistochemistry figures (Figures 1–5) are visually clear but the legends are very descriptive without specifying magnification or scale bars. Please include, for each panel, the objective magnification (e.g. “H&E, ×40/×100/×200”) and/or a scale bar. Ensure that all panel labels are explicitly described in the legend so the reader can easily map.

The stains (HE or the specific immunohistochemistry) are now displayed in the lower left corner of each image. The original magnifications has been added at the end of the legend.

The Abbreviations list is helpful; please check that all abbreviations (e.g., AChR, MuSK, MALT) are defined at first use in the text and align exactly with the list at the end.

The abbreviation list has been updated and a consequent definition of each of them upon first use in the text das been done avoiding the use of abbreviations in the Simple summary, Abstract and Figure legends.

Reviewer 2 Report

Comments and Suggestions for Authors

Dear author, thank you for giving me the opportunity to review your article entitled "Thymic Hyperplasias in Practice: Clinical Context, Histological Clues, and Management Implications". I have some comments and suggestions:

  • I think you could incorporate a paragraph on radiological imaging (aspect on CT, MRI and PET-CT) for each entities.
  • Old references: Thymus weight relies on tables from 1927 and 1931.. Do you have any data on CT volumetric criteria. Please incorporate or reference modern radiological literature regarding normal thymic thickness and volume by age if possible.
  • As clinician, I would be interested to know when it is necessary to make a biopsy . Can you clarify this point in the discussion.
  • You mentionned that 14% of LESA-like TH patients had an associated lymphoma, most commonly MALT lymphoma. What is the current management for these patients?

Author Response

Reviewer #2

Dear author, thank you for giving me the opportunity to review your article entitled "Thymic Hyperplasias in Practice: Clinical Context, Histological Clues, and Management Implications". I have some comments and suggestions:

I think you could incorporate a paragraph on radiological imaging (aspect on CT, MRI and PET-CT) for each entities.

Thank you for this suggestion. Wherever possible, these aspects have been included; however, the radiologic findings remain largely non-specific. Similar to the challenges encountered in histopathology, for certain entities such as true thymic hyperplasia and LESA-like thymic hyperplasia, extensive experience and systematic studies are still lacking. Interestingly, a very recent publication reported increased FDG avidity in a patient with LESA-like thymic hyperplasia.

Old references: Thymus weight relies on tables from 1927 and 1931.. Do you have any data on CT volumetric criteria. Please incorporate or reference modern radiological literature regarding normal thymic thickness and volume by age if possible.

In the revised manuscript, I provide age-stratified CT reference values for thymic dimensions in children and adolescents. Previous studies by Colak et al. and Araki et al. have reported detailed analyses of thymic parameters, though adult diameters reported by Araki et al. are larger than those in children, highlighting the need for precise and standardized measurement. Therefore, thymic assessment should not rely on size alone; a multiparametric imaging approach incorporating tissue composition, CT attenuation, internal morphology, enhancement patterns, and correlation with age and clinical context, with ultrasound and MRI as complementary modalities, is recommended.

As clinician, I would be interested to know when it is necessary to make a biopsy . Can you clarify this point in the discussion.

Thank you for this comment. The challenges associated with biopsies are specific to each of the four entities, and I have supplemented the considerations for a biopsy in  the respective sections. For example, in true thymic hyperplasia (TFH), many patients already have a known and treated myasthenia gravis, so a biopsy is unlikely to contribute diagnostically and may yield little or only atrophic thymic tissue. In LESA-like thymic hyperplasia, crucial histological features or an emerging lymphoma could be missed. Therefore, the indication for biopsy in these entities should be considered with caution, although it may be justified in certain situations, such as in patients with high perioperative risk or relevant radiologic differential diagnoses.

You mentionned that 14% of LESA-like TH patients had an associated lymphoma, most commonly MALT lymphoma. What is the current management for these patients?

Experience with LESA-like thymic hyperplasia, particularly regarding associated lymphomas, is very limited and, in my opinion, does not allow firm conclusions beyond the following: (1) Pathologists should be aware of this association and systematically evaluate for it in tissue sections; (2) Upon diagnosis - as with any lymphoma - patients should be referred to a hematologist for appropriate workup, including staging.

Reviewer 3 Report

Comments and Suggestions for Authors

The manuscript offers a comprehensive, articulate, and technically competent overview of the heterogeneous group of thymic hyperplasias, presenting the topic with a clarity that makes the text accessible both to specialists in thoracic pathology and to clinicians who manage mediastinal lesions. One of the most appreciable aspects of the article is its ability to synthesize a large body of scattered literature and integrate it with practical, case-based diagnostic reasoning, particularly in settings where small biopsies complicate morphological interpretation. The author consistently interweaves clinical, serological, histopathological and radiologic information, and this multidimensional approach greatly enhances the clinical relevance of the review. The distinctions drawn between true thymic hyperplasia, rebound hyperplasia, thymic follicular hyperplasia and LESA-like thymic hyperplasia are explained with precision and nuance, avoiding schematic simplifications while remaining entirely comprehensible.

From a scientific standpoint, the discussion around diagnostic pitfalls is one of the strongest contributions of the manuscript. The careful differentiation between hyperplastic processes and B1 thymomas, micronodular thymomas with lymphoid stroma, T-lymphoblastic lymphoma and MALT lymphoma reflects a solid understanding of real-world diagnostic challenges. The emphasis placed on architectural preservation, cortical–medullary relationships, follicular distribution and immunohistochemical markers gives the article a clear practical utility. The section on LESA-like thymic hyperplasia is particularly valuable, especially in highlighting its non-myasthenic autoimmune associations and the non-negligible frequency of concomitant MALT lymphomas. This reinforces the concept that LESA-like TH is not simply a hyperplastic variant of TFH but a distinct pathological entity with implications that extend beyond morphology.

Although the manuscript is strong overall, the methodology could be presented with greater transparency. The description of the Medline search strategy is extremely general and does not specify how sources were selected or prioritised. While narrative reviews are not required to follow systematic protocols, even a brief account of inclusion criteria or the rationale behind the incorporation of very old thymus-weight literature would improve reproducibility and methodological credibility. In addition, the review acknowledges the heterogeneity and limitations of early 20th-century studies on thymus weight, yet it does not fully explore how these limitations affect contemporary diagnostic boundaries or normative thresholds. A more explicit critical appraisal of the reliability of historical data would be welcome, particularly given the reliance on these values in distinguishing true hyperplasia from normal variation.

A further area that could be strengthened is the absence of molecular considerations. While the field is still limited in this regard, even a short discussion about whether any emerging molecular markers, clonality studies or pathways have been explored in LESA-like hyperplasia or TFH would position the review more firmly within contemporary diagnostic pathology. This absence is not a flaw per se, but the article would benefit from a brief statement clarifying the current state of molecular evidence.

One aspect that could be improved from a didactic perspective is the inclusion of a diagnostic flowchart or a more schematic conceptual synthesis. The text is rich and dense, and although Table 4 provides a helpful summary, a visual algorithm highlighting discriminating features such as architectural preservation, follicular distribution, epithelial hyperplasia, cortical integrity and clinical associations would significantly enhance its utility, especially for pathology trainees. While the prose is highly organised, a visual complement would transform the manuscript into a more practical reference tool.

Despite these considerations, the article remains an excellent and rigorous contribution. It succeeds in clarifying a topic that is often misunderstood due to the overlapping morphology of thymic lesions and the rarity of some entities. It provides a convincing argument for the integration of clinical history, serology and imaging into pathologic interpretation, repeatedly demonstrating that morphology alone is not sufficient to accurately classify thymic hyperplasias. The discussion of LESA-like thymic hyperplasia in particular stands out as a segment that will likely serve as a useful reference for clinicians and pathologists confronted with unusual mediastinal biopsies. With minor refinements in methodological exposition and the addition of a synthesizing visual element, the manuscript has the potential to function not only as an informative review but as an authoritative practical guide within the field.

Author Response

Reviewer #3

The manuscript offers a comprehensive, articulate, and technically competent overview of the heterogeneous group of thymic hyperplasias, presenting the topic with a clarity that makes the text accessible both to specialists in thoracic pathology and to clinicians who manage mediastinal lesions. One of the most appreciable aspects of the article is its ability to synthesize a large body of scattered literature and integrate it with practical, case-based diagnostic reasoning, particularly in settings where small biopsies complicate morphological interpretation. The author consistently interweaves clinical, serological, histopathological and radiologic information, and this multidimensional approach greatly enhances the clinical relevance of the review. The distinctions drawn between true thymic hyperplasia, rebound hyperplasia, thymic follicular hyperplasia and LESA-like thymic hyperplasia are explained with precision and nuance, avoiding schematic simplifications while remaining entirely comprehensible. From a scientific standpoint, the discussion around diagnostic pitfalls is one of the strongest contributions of the manuscript. The careful differentiation between hyperplastic processes and B1 thymomas, micronodular thymomas with lymphoid stroma, T-lymphoblastic lymphoma and MALT lymphoma reflects a solid understanding of real-world diagnostic challenges. The emphasis placed on architectural preservation, cortical–medullary relationships, follicular distribution and immunohistochemical markers gives the article a clear practical utility. The section on LESA-like thymic hyperplasia is particularly valuable, especially in highlighting its non-myasthenic autoimmune associations and the non-negligible frequency of concomitant MALT lymphomas. This reinforces the concept that LESA-like TH is not simply a hyperplastic variant of TFH but a distinct pathological entity with implications that extend beyond morphology.

Although the manuscript is strong overall, the methodology could be presented with greater transparency. The description of the Medline search strategy is extremely general and does not specify how sources were selected or prioritised. While narrative reviews are not required to follow systematic protocols, even a brief account of inclusion criteria or the rationale behind the incorporation of very old thymus-weight literature would improve reproducibility and methodological credibility. In addition, the review acknowledges the heterogeneity and limitations of early 20th-century studies on thymus weight, yet it does not fully explore how these limitations affect contemporary diagnostic boundaries or normative thresholds. A more explicit critical appraisal of the reliability of historical data would be welcome, particularly given the reliance on these values in distinguishing true hyperplasia from normal variation.

Thank you for this comment. In the revised manuscript, I have supplemented the tables with additional data from the publications by Hammar et al. and Greenwood and Woods, using the same age groups, which allows for substantially higher sample sizes per group. I acknowledge that all three studies are older; however, their similar design permits meaningful aggregation of the data. More importantly, as suggested by Reviewer #1 and #2, information on radiologic findings has been included. I refer the reader to the studies by Colak et al. and Araki et al., which provide detailed age-related descriptions of multiple radiologic parameters.

A further area that could be strengthened is the absence of molecular considerations. While the field is still limited in this regard, even a short discussion about whether any emerging molecular markers, clonality studies or pathways have been explored in LESA-like hyperplasia or TFH would position the review more firmly within contemporary diagnostic pathology. This absence is not a flaw per se, but the article would benefit from a brief statement clarifying the current state of molecular evidence.

Among these four entities, the only one for which some pathogenetic insights are available is thymic follicular hyperplasia (TFH), and I have included a brief summary of this information. Unfortunately, for true thymic hyperplasia (TTH), rebound thymic hyperplasia (RTH), and LESA-like thymic hyperplasia, no specific molecular mechanisms have been identified, aside from the described associations. A brief note on this, including a statement on the polyclonality of the lymphoid expansion, has been added.

One aspect that could be improved from a didactic perspective is the inclusion of a diagnostic flowchart or a more schematic conceptual synthesis. The text is rich and dense, and although Table 4 provides a helpful summary, a visual algorithm highlighting discriminating features such as architectural preservation, follicular distribution, epithelial hyperplasia, cortical integrity and clinical associations would significantly enhance its utility, especially for pathology trainees. While the prose is highly organised, a visual complement would transform the manuscript into a more practical reference tool.

Thank you for this suggestion. I have now designed a visual overview that conceptualizes histological changes and integrates them with the clinical information in an algorithm-like approach (Figure 6).

Despite these considerations, the article remains an excellent and rigorous contribution. It succeeds in clarifying a topic that is often misunderstood due to the overlapping morphology of thymic lesions and the rarity of some entities. It provides a convincing argument for the integration of clinical history, serology and imaging into pathologic interpretation, repeatedly demonstrating that morphology alone is not sufficient to accurately classify thymic hyperplasias. The discussion of LESA-like thymic hyperplasia in particular stands out as a segment that will likely serve as a useful reference for clinicians and pathologists confronted with unusual mediastinal biopsies. With minor refinements in methodological exposition and the addition of a synthesizing visual element, the manuscript has the potential to function not only as an informative review but as an authoritative practical guide within the field.

Round 2

Reviewer 2 Report

Comments and Suggestions for Authors

Dear author, thank you for your responses. The manuscript is fine for me.